Do syntopic host species harbour similar symbiotic communities? The case of Chaetopterus spp. (Annelida: Chaetopteridae)

Britayev Temir A. 1
Mekhova Elena 1
Deart Yury 1
Martin Daniel dani@ceab.csic.es 2
1 Severtzov Institute of Ecology and Evolution, Russian Academy of Sciences , Moscow , Russian Federation
2 Department of Marine Ecology, Centre d’Estudis Avançats de Blanes (CEAB–CSIC) , Blanes , Catalunya , Spain
Reimer James
Electronic publication date: 2017 Feb 2
Publication date: 2017
Volume: 5
Electronic Location ID: e2930
Received 2016 Oct 10; Accepted 2016 Dec 21
Copyright: ©2017 Britayev et al.
Copyright year: 2017
Copyright holder: Britayev et al.
License: This is an open access article distributed under the terms of the Creative Commons Attribution License, which permits unrestricted use, distribution, reproduction and adaptation in any medium and for any purpose provided that it is properly attributed. For attribution, the original author(s), title, publication source (PeerJ) and either DOI or URL of the article must be cited.
License URL: https://creativecommons.org/licenses/by/4.0/

Keywords: Symbiotic community structure, Polychaeta, Carapidae, Porcellanidae, Pinnotheridae, Tergipedidae, Competition, South China Sea, Vietnam

Funding: Russian Science Foundation 14-14-01179 Spanish Research Project MariSymBiomics CTM2013-43287-P Consolidated Research Group of Marine Benthic Ecology of the Generalitat de Catalunya 2014SGR120 This work was suported by the Russian Science Foundation (grant no 14-14-01179), the Spanish Research Project MariSymBiomics (CTM2013-43287-P) and the Consolidated Research Group of Marine Benthic Ecology of the Generalitat de Catalunya (2014SGR120). The funders had no role in study design, data collection and analysis, decision to publish, or preparation of the manuscript.

==============================
To assess whether closely related host species harbour similar symbiotic communities, we studied two polychaetes, Chaetopterus sp. (n = 11) and Chaetopterus cf. appendiculatus (n = 83) living in soft sediments of Nhatrang Bay (South China Sea, Vietnam). The former harboured the porcellanid crabs Polyonyx cf. heok and Polyonyx sp., the pinnotherid crab Tetrias sp. and the tergipedid nudibranch Phestilla sp. The latter harboured the polynoid polychaete Ophthalmonoe pettiboneae, the carapid fish Onuxodon fowleri and the porcellanid crab Eulenaios cometes, all of which, except O. fowleri, seemed to be specialized symbionts. The species richness and mean intensity of the symbionts were higher in Chaetopterus sp. than in C. cf. appendiculatus (1.8 and 1.02 species and 3.0 and 1.05 individuals per host respectively). We suggest that the lower density of Chaetopterus sp. may explain the higher number of associated symbionts observed, as well as the 100% prevalence (69.5% in C. cf. appenciculatus). Most Chaetopterus sp. harboured two symbiotic species, which was extremely rare in C. cf. appendiculatus, suggesting lower interspecific interactions in the former. The crab and nudibranch symbionts of Chaetopterus sp. often shared a host and lived in pairs, thus partitioning resources. This led to the species coexisting in the tubes of Chaetopterus sp., establishing a tightly packed community, indicating high species richness and mean intensity, together with a low species dominance. In contrast, the aggressive, strictly territorial species associated with C. cf. appendiculatus established a symbiotic community strongly dominated by single species and, thus, low species richness and mean intensity. Therefore, we suggest that interspecific interactions are determining species richness, intensity and dominance, while intraspecific interactions are influencing only intensity and abundance. It is possible that species composition may have influenced the differences in community structure observed. We hypothesize that both host species could originally be allopatric. The evolutionary specialization of the symbiotic communities would occur in separated geographical areas, while the posterior disappearance of the existing geographical barriers would lead to the overlapped distribution.

Introduction

During the last few decades, significant efforts have been undertaken to study the species composition and structure of marine symbiotic communities associated with different hosts taxa such as scleractinian corals (Hoeksema, Van der Meij & Fransen, 2012; Stella, Jones & Pratchett, 2010), echiurans (Anker et al., 2005), hermit crabs (Williams & McDermott, 2004) and echinoderms (Barel & Kramers, 1977). Despite this being an interesting aspect of marine ecosystems’ functioning and the need to fill in existing gaps in related knowledge, the current focus of scientific interests have shifted to ecological and evolutionary aspects of the establishment of symbiotic communities (Baeza, 2015; Duffy, 2002; Thiel & Baeza, 2001). Accordingly, host characteristics (morphological, ecological and physiological) have been considered as some of the most important parameters driving these processes (e.g., Abele & Patton, 1976; Deheyn, Lyskin & Eeckahaut, 2006; Goto & Kato, 2011).

The coexistence of potential hosts that are taxonomically closely related (thus sharing similar morphological and physiological characteristics) may facilitate host switching, leading to the infestation of different host species by the same species of symbiont, as reported for example in freshwater fish (Poulin, 1998). Accordingly, we may expect the composition of symbiotic communities established on closely related hosts to be similar. Hence, sympatric coral species belonging to the same family harbour symbiotic communities more similar than those belonging to different families (Stella, Jones & Pratchett, 2010), while the symbiotic communities associated with two starfish hosts from the same family living in the same area have nearly identical species composition (Antokhina, Savinkin & Britayev, 2012). There seems to be a correlation between increasing taxonomic proximity between hosts and a higher similarity in species composition of the respective symbiotic communities. In other words, we could expect that closely related (i.e., belonging to the same genus) host species sharing the same habitat would harbour very similar (or even identical) symbiotic communities. Therefore, the current study investigated the symbiotic communities associated with two species of Chaetopterus in Nhatrang Bay (Vietnam), to assess whether this hypothesis may apply to this particular situation.

These two species of Chaetopterus appeared to be excellent subjects for the intended comparison due to their highly similar morphology. In fact, the genus has long been regarded as monospecific and, to date, the morphological identification of species is still considered as rather complex (Britayev & Martin, 2016; Nishi, Hickman Jr & Bailey-Brock, 2009; Petersen, 1984a; Petersen, 1984b). Moreover, these two species share the same habitat and, thus the influence of environmental parameters can be excluded as influential factors on the associated symbiotic communities.

The genus Chaetopterus (Annelida: Chaetopteridae) includes relatively large animals (up to 20–25 cm in length) living in roughly U-shaped tubes embedded into soft sediments or attached to hard surfaces in shallow waters of temperate and tropical seas (Britayev & Martin, 2016). Morphologically, they are highly adapted for feeding on plankton using complex mucus-net based mechanisms (Enders, 1909). They are also well known as hosts harbouring numerous symbiotic associates (often including complex communities) inside their parchment-like tubes. These tubes provide well-protected shelter with continuous water flow bringing oxygen and food items to the symbionts (Britayev & Martin, 2016). To date, approximately 28 species of symbionts have been reported living inside tubes of Chaetopterus (Petersen & Britayev, 1997). However, information on the composition of associated communities is lacking, and is currently only available for two species, C. pergamentaceus Cuvier, 1830 and C. cf. cautus Marenzeller, 1879, which are each host to 3–5 species of crabs and polychaetes (Britayev, 1993; Gray, 1961).

A species of Chaetopterus (not confirmed but probably Chaetopterus cf. appendiculatus Grube, 1874) inhabiting Vietnamese soft seabed sediments was previously reported as harbouring three species of symbionts within its tubes: the polychaete Ophthalmonoe pettiboneae Petersen & Britayev, 1997, an unidentified carapid fish and a porcellanid crab (Britayev & Martin, 2005). The presence of a second, probably undescribed, species of Chaetopterus sharing the same habitat and having its own associated symbiotic community allowed us to investigate the hypothesis that postulates the similarity in composition of symbiotic communities associated with morphologically similar hosts.

More specifically, in this paper we analyse: (1) The morphological and ecological characteristics of the two Vietnamese host species of Chaetopterus; (2) The composition, species richness and abundance of the symbiotic communities associated with the two host species; and (3) The host specificity of all symbiotic species.

Material and Methods

Sampling was conducted between March and April 2016 in four localities of Nhatrang Bay (Vietnam, South China Sea): the western coast of Mun Island, the southern coast of Mot Island, the western coast of Tre Island and Dam Bay (Fig. 1, Table 1). The Russian-Vietnamese Tropical Center issued a letter supporting the collection of samples and animals used in this paper.

Figure 1 (A) Location of the study area on the Vietnamese coast of the South China Sea. (B) Sampling sites, Nhatrang Bay. Map data: SIO, NOAA, US Navy, NGA, GEBCO. Image (C) 2016 Digital Globe and Google Earth.

Table 1 Depth (m) and geographical coordinates of the studied locations, as well as density (chaetopterid individuals per 100 m2/per diving hour), ratio of individuals (Chaetopterus sp. vs. Chaetopterus cf. appendiculatus) and infestation prevalence (%) of the respective host populations

Locality	Station	Latitude	Longitude	Depth	Density	Ratio	Prevalence	
Mun Island	1	12°10′10″N	109°17′46″E	13–16	1.0/3.8	2/20	65.2	
Dam Bay	2	12°11′43″N	109°17′26″E	6–8	0.6/3.0	2/23	81.8	
Mot Island	3	12°10′26″N	109°16′23″E	16–20	nd/4.7	7/20	90.9	
Point Nam Tre Island	4	12°13′42″N	109°13′47″E	10–12	nd/7.1	1/19	60.0	
Notes.

nd no data

The chaetopterid hosts were collected by SCUBA diving at depths from 6 to 20 m. As their tubes were embedded within the sediment to 15–20 cm depth, extraction was achieved by washing out the sediments by hand. The tubes were then gently removed, immediately placed into individual zip-lock plastic bags to avoid losing symbionts and transferred to seawater tanks, where they were kept until reaching the laboratory facilities.

The density of the studied population of Chaetopterus was estimated along five 50 m long and 2 m wide transects at Mun Island and Dam Bay. At each site, the transects followed the depth profile and were placed parallel, each one immediately adjacent to the previous one. Two divers were responsible for counting the number of chaetopterid tube siphons, each one on one side (1 m) of the transect. A second density estimate was based on the number of chaetopterids sampled per hour at each sampling site (except when diving surveys were used for transect estimates).

In the laboratory, tube length was measured to the nearest 5.0 mm (Table S1). Then, tubes were gently opened by hand and carefully checked for presence of symbionts. The species and number of symbionts were recorded (Table S1). Water and sediment from the bag were sieved through a 1 mm mesh and the retained sediments were carefully inspected by eye. The body in Chaetopterus is divided into three differentiated regions: the nine-to-ten anterior-most segments, the five mid-body segments, and an undefined (but usually very numerous) number of posterior segments, which form the regions A, B, and C, respectively (Britayev & Martin, 2016). Hosts were extracted and measured either as length and width for region A (n = 8) or as displaced water volume in a graduate vessel to the nearest 1 ml (all remaining specimens) (Table S1). As body volume showed a positive linear relationship with tube length (Tube length = 44.084 + 0.503*Body volume, F = 26.457, P < 0.0001; Table S2), this easy-to-obtain measurement was used to study community structure.

All chaetopterid tubes, hosts and symbionts were photographed with Canon digital cameras (G16 and EOS 6D). Selected hosts and all symbionts were fixed either in 70% or 99% ethanol, or in a 4% formaldehyde/seawater solution for further studies. Small fragments of the ventral uncinal tori of both host species have been dissected. To illustrate the shape of the uncini, these fragments were squashed between slides, mounted in glycerine and photographed with the help of a ProgRes C10 Plus digital camera (Jenoptics, Jena) attached to a Zeiss Axioplan compound microscope.

All symbionts were measured to the nearest 0.1 mm, using a calibrated ocular micrometer under an Olympus SZX9 stereomicroscope as body length from tip of prostomium to the end of pygidium for polychaetes, as body length from tip of head to the end of caudal peduncle for fish, and as carapace width for crabs. Crabs were sexed according to the abdominal shape.

DNA was extracted using Spin Columns Thermo Scientific GeneJET 50 kit, following the manufacturer’s standard protocol. 10 ng of genomic DNA was used as a template for polymerase chain reaction (PCR) with special mitochondrial Cox1 primers: TGTAAAACGACGGCCAGTGAYTATWTTCAACAAATCATAAAGATATTGG and CAGGAAACAGCTATGACTAMACTTCWGGGTGACCAAARAATCA (Carr et al., 2011). PCR were set up in total volume of 20 µl. The PCR cycling profiles were as follows: initial denaturation (95 °C, 5 min); followed by 35 cycles of denaturation (95 °C, 15 s), annealing (45 °C, 15 s) and extension (72 °C, 60 s). The resulting PCR products were purified by direct purification from the PCR mixture and prepared for sequencing. Overlapping sequence fragments were merged into consensus sequences using MEGA7 (Kumar, Stecher & Tamura, 2016), the protein coding COI being simple to align. The obtained COI sequences and voucher paragenophores (Pleijel et al., 2008) for the two species of Chaetopterus have been deposited in GenBank and in the collections of the Severtsov Institute of Ecology and Evolution RAS, respectively. Seven host specimens were used in genetic analyses to ensure species delineation (Table 2). The genetic differentiation within and between species was assessed by pairwise genetic distances between COI sequences using the Maximum Likelihood Model, which allowed us to show the percentage of replicate trees in which the associated taxa clustered together in the bootstrap test (1,000 replicates) next to each branch (Felsenstein, 1985). The phylogenetic tree was drawn to scale, with branch lengths in the same units as those of the evolutionary distances used to infer it, as computed using the Maximum Composite Likelihood method (Tamura, Nei & Kumar, 2004) and are in the units of the number of base substitutions per site. The phylogenetic tree was built using the COI sequences of Chaetopterus and Mesochaetopterus available from NCBI GenBank, using Spiochaetopterus costarum (Claparéde, 1869) as the outgroup, by means of the Neighbour-Joining method (Saitou & Nei, 1987) in MEGA7 (Kumar, Stecher & Tamura, 2016).

Table 2 Specimens list for the two Vietnamese host species used in the molecular analyses, detailing the GenBank accession numbers and the collection references for the voucher paragenophores.

Chaetoperus	Specimen number	Accession number	Voucher	
cf. appendiculatus	14	KY124465	sevin Pl/Vn 2016Ch0001	
cf. appendiculatus	76	KY124466	sevin Pl/Vn 2016Ch0002	
cf. appendiculatus	77	KY124467	sevin Pl/Vn 2016Ch0003	
cf. appendiculatus	80	KY124468	sevin Pl/Vn 2016Ch0004	
sp.	16	KY124469	sevin Pl/Vn 2016Ch0005	
sp.	82	KY124470	sevin Pl/Vn 2016Ch0006	
sp.	93	KY124471	sevin Pl/Vn 2016Ch0007	

For the purposes of our study, the following terms are defined: Prevalence, as the ratio between number of infested and total number of hosts; Intensity, as the number of symbionts present in each infested host; Mean intensity, as the mean number of individuals of a particular symbiotic species per infested host in a sample; Abundance, as mean number of symbionts per examined host, infested and non-infested; and Species richness, as mean number of symbiotic species per infested host.

The porcellanid crabs were identified by Prof. Bernd Werding, from the Institut für Allgemenie und Spezielle Zoologie of the Justus-Liebig Universität (Giessen, Germany). The pinnotherid crab was identified by Prof. Peter Ng from the Department of Zoology of the National University of Singapore (Republic of Singapore). The carapid fish was identified by Dr. Eric Parmentier from the Laboratoire de Morphologie Fonctionnelle et Evolutive of the Institut de Chimie of the Université de Liege (Belgium). The tergipedid nudibranch was identified by Dr. Irina Ekimova, from the Department of Invertebrate Zoology of the Lomonosov Moscow State University (Russian Federation).

The relationship between host body volume and tube length were assessed by linear regression. The species richness and mean intensity, as well as the average length of infested and non-infested tubes of Chaetopterus, were compared by Student’s t-test. Statistical analyses were performed using Statistica 6.0 and PAST 2.17 software.

Results

Hosts characteristics

The two Vietnamese host species of Chaetopterus are morphologically similar. However, one of them is significantly bigger than the other, both in terms of tube length (1.4:1, on average) and body volume (2.7:1, on average) (t-test p < 0.0001, Table 3). They also differ in the number of chaetigers of region A (9 and 9–11, respectively) (Figs. 2A and 2D) and in the denticles of the neuropodial uncini of region C (25–35 and 9, respectively) (Figs. 3A–3D), as well as in tube structure. Tubes of the bigger species are covered by silt, have a parchment-like appearance and the inner lining is iridescent, silver or golden in colour, showing distinct transverse annulations (Fig. 2C). In the smaller species, tubes are covered by sand and small coral and shell fragments, have a paper-like appearance with a semi-transparent inner lining, whitish or brownish in colour and lacking distinct annulations (Fig. 2F).

Table 3 Number of individuals, mean tube length (min–max), cm and mean body volume (min–max) of Chaetopterus cf. appendiculatus and Chaetopterus sp.

Species	Number	Tube length (cm)	Body volume (cm2)	
Chaetopterus cf. appendiculatus	83	64.6 (41–88)	41.9 (23–72)	
Chaetopterus sp.	11	44.8 (23–58)	15.8(2–32)	

Figure 2 Chaetopterus cf. appendiculatus: (A) whole worm; (B) tube; (C) detail of inner tube surface. Chaetopterus sp.: (D) whole worm; (E) tube; (F) detail of inner tube surface.

Scale bars are 5 cm.

Figure 3 Uncini from ventral neuropodial tori of region C.

Chaetopterus cf. appendiculatus: (A) upper tori; (B) lower tori. Chaetopterus sp.: (C) upper tori; (D) lower tori.

We tentatively identified the bigger host species as Chaetopterus cf. appendiculatus because, according to the original description, this species has a sand-coloured inner tube surface, showing dense transverse annulations. Additionally, it is the only described species of Chaetopterus possessing neuropodial uncini from region C with more than 20 small denticles. Chaetopterus appendiculatus was already reported as host of O. pettiboneae from the Banda Sea (Indonesia) by Petersen & Britayev (1997). Petersen (1997) proposed the redescription of C. appendiculatus as a valid species, based on the type material from Ceylon. The fact that formal redescription has never been published does not prevent us from considering the species as valid, whose formal redescription is far beyond the scope of this paper. However, the long geographical distance between Ceylon/Indonesia and Vietnam prevents us in fully assigning the Vietnamese specimens to C. appendiculatus, and we refer to the species as C. cf. appendiculatus in this paper. The smaller host is likely  undescribed.

The phylogenetic analysis including the COI sequences of the Vietnamese hosts (Fig.  4) showed low bootstrap values that did not allow us to fully resolve the phylogeny of Chaetopterus. However, it clearly revealed that the two Vietnamese hosts are different species included within two separate monophyletic clades (with 100% bootstrap support), thus confirming our morphological inference. Although with low support, the closest clades to those of the two Vietnamese Chaetopterus belong to C. variopedatus (Renier, 1804). However, the specimens joining the Chaetopterus sp. clade (42% bootstrap support) originate from the Mediterranean, while those joining the C. cf. appendiculatus clade (54% bootstrap support) originate from the Atlantic. As indicated by Martin et al. (2008), our results support the inference that the two populations of C. variopedatus belong to different species, with the Mediterranean species described and the Atlantic species still undescribed. The results also confirm that C. variopedatus sensu Hartman (1959) is not a single cosmopolitan species, but a complex including more than 20 different species (Bhaud, 1998; Osborn et al., 2007; Petersen, 1984a; Petersen, 1984b; Petersen, 1997). As is the case for C. appendiculatus, some of these species have not yet been formally redescribed. However, as many as nine species have recently been described, and five have been redescribed in the recent literature (Nishi, 2001; Nishi, Arai & Sasanuma , 2000; Nishi, Hickman Jr & Bailey-Brock, 2009; Osborn et al., 2007; Sun & Qiu, 2014).

Figure 4 Preliminary phylogenetic tree for species of Chaetopterus and Mesochaetopterus based on the COI sequences obtained from NCBI GenBank and our data.

The sequences for the two Vietnamese species are listed in Table 2.

The two Vietnamese Chaetopterus host species were found at the same localities, with their tubes deeply embedded in silty sand sediments. Chaetopterus cf. appendiculatus outnumbered Chaetopterus sp. in all samples, yet their proportion varied depending on the locality, with Chaetopterus sp. being relatively more abundant at Mot Islands (St. 2, 25.9%) and substantially less abundant at the other stations (5.0–9.1%) (Table 1).

The density of Chaetopterus ranged from 0.6 to 1.0 individuals per 100 m2 in the transects, while the number of collected worms per diving hour was lower at St. 4 in Dam Bay and higher at St. 1 in Point Nam, Tre Island (Table 1).

Taxonomic composition of the symbiotic communities

91 individuals of seven species of animals occurred in association with the two host species of Chaetopterus. Among them, the polynoid polychaete Ophthalmonoe pettiboneae (Fig. 5C), the tergipedid nudibranch Phestilla sp. (Fig. 6G), the carapid fish Onuxodon fowleri (Smith, 1964) (Fig. 5D), and four species of decapods, three porcellanids, Eulenaios cometes (Walker, 1887) (Figs. 5A and 5B), Polyonyx cf. heok (Osawa & Ng, 2016) (Figs. 6A and 6B) and Polyonyx sp. (Figs. 6E and 6F), and the pinnotherid Tetrias sp. (Figs. 6C and 6D) (Table 4).

Figure 5 Symbiotic species associated with Chaetopterus cf. appendiculatus: (A, B) Eulenaios cometes (female and male, respectively); (C) Ophthalmonoe pettibonneae; (D) Onuxodon fowleri.

Scale bars are 1  cm.

Figure 6 Symbiotic species associated with Chaetopterus sp.: (A, B) Polyonyx cf. heok (male and female, respectively); (C, D) Tetrias sp. (male and female, respectively); (E, F) Polyonyx sp., (male and female, respectively); (G, F) Phestilla sp. (whole body and egg-mass, respectively); egg-mass indicated by arrows.

Scale bars are 0.5 cm.

Table 4 Prevalence (%) and mean intensity (mean number of individuals per infested host) of the symbiotic species associated with the two host Chaetopterus.

Symbiont species	Chaetopterus sp.	Chaetopterus cf. appendiculatus	
Ophthalmonoe pettiboneae (P)	–	64.1 (1.0)	
Phestilla sp. (G)	22.2 (2.0)	–	
Eulenaios cometes (D)	–	1.3 (2.0)	
Polyonyx cf. heok (D)	88.9 (1.7)	–	
Polyonyx sp. (D)	66.7 (1.3)	–	
Tetrias sp. (D)	11.1 (2.0)	–	
Onuxodon fowleri (A)	–	6.4 (1.2)	
Notes.

P Polychaeta

D Decapoda

G Gastropoda

A Actinopteri

Four and three species were found inside the tubes of Chaetopterus sp. and C. cf. appendiculatus, respectively. Surprisingly, the symbiotic communities associated with the two hosts did not have any species in common, with the only similarity at a higher taxonomic level being the presence of porcellanid crabs (Table 4). Despite the lower sample size of Chaetopterus sp., the diversity of its associated community was higher than that of C. cf. appendiculatus. Accordingly, it may be expected that the number of species associated with Chaetopterus sp. would increase with an increasing number of analysed host individuals. Conversely, the diversity of the community associated with C. cf. appendiculatus showed an almost saturated species accumulation curve (Fig. 7A).

Figure 7 Characterization of the symbiotic communities associated with the two host species of Chaetopterus: (A) rarefaction curve; (B) distribution of symbionts per host; (C) relative abundance of the symbiotic species.

Ophthlmonoe pettiboneae is the single symbiotic species previously known from Vietnamese waters and from the same host species. The other six are here reported for the first time from the Vietnamese coasts. Moreover, O. fowleri is herein reported as a symbiont of chaetopterids for the first time, as well as Tetrias sp., Polyonyx sp., P. cf. heok and Phestilla sp., which are new to science and will be described at later dates in specialized papers. The tergipedid nudibranch is also, to the best of our knowledge, the first known nudibranch living in symbiosis with a polychaete host. It shows a posterior end functioning as a sucker (Fig. 6F) allowing it to attach to the smooth inner surface of the host tube, while the rest of the body can move freely (https://www.researchgate.net/publication/310159685_Phestilla_sp). Its flattened body, together with the lack of cnidosacs and the uniserial radula with long lateral denticles on the rachidian tooth clearly place it within the genus Phestilla. However, it differs from all known species of this genus by having a small central denticle of the radula, a wider foot and cerata arranged one per row only (Y Deart & I Ekimova, pers. comm., 2016). Moreover, its appearance and colouring (Fig. 6F) mimics, to some extent, that of the very posterior end of the chaetopterid host.

Structure of the symbiotic communities

A total of 61 (73.5%) out of 83 individuals of C. appediculatus and all 11 (100%) Chaetopterus sp. were infested. Among the species associated with C. appediculatus, O. pettiboneae showed a higher prevalence than the two other symbionts (Table 4). Among the associates with Chaetopterus sp., the prevalence ranged from 11% to almost 90%, with the maximum corresponding to P. cf. heok (Table 4).

The number of species inhabiting the same tube varied from zero to two in C. cf. appendiculatus, and from one to two in Chaetopterus sp. However, the species richness was significantly higher in the latter (p > 0.001, Figs. 7A and 7B) due to the common coexistence of two symbiont species in the same host tube. In fact, the small-sized Polyonyx sp. and Phestilla sp. (Figs. 6E and 6G) were found in all observed cases living with other crab species, mostly with the large sized P. cf. heok (Figs. 6A and 6B). Only in one case, Polyonyx sp. shared the host tube with Tetrias sp. (Table 5). In contrast, most tubes of C. appediculatus were occupied by a single symbiotic species, either O. pettiboneae or O. fowleri. Only in one case two species of symbionts (O. pettiboneae and E. cometes) were present sharing the same host tube (Table 5).

Table 5 Distribution among hosts (as number of host tubes found without and with one, two and three individuals) for the five species associated with Chaetopterus spp.

Symbiont	Chaetopterus	0	1	2	3	4	
Ophthalmonoe pettiboneae	Ca	21	56	0	0	0	
Onuxodon fowleri	Ca	78	5	1	0	0	
Eulenaios cometes	Ca	8	0	1	0	0	
Polyonyx cf. heok	Cs	1	3	7	0	0	
Polyonyx sp.	Cs	3	4	2	0	0	
Tetrias sp.	Cs	10	0	1	0	0	
Phestilla sp.	Cs	8	1	1	1	0	
Notes.

Cs Chaetopterus sp.

Ca Chaetopterus cf. appendiculatus

The number of symbiont individuals infesting one host varied from zero to three in C. cf. appediculatus, and from one to five in Chaetopterus sp., while the mean intensity was nearly three times lower in the former than in the later (p > 0.001, Table 6). Accordingly, most C. cf. appediculatus were infested by one symbiotic individual, while multiple infestation (two, three, or even four symbionts) was common in Chaetopterus sp. (Fig. 7B).

Table 6 Symbiotic community indexes: species richness (mean number of species per one infested host), infestation prevalence (%), and mean intensity (mean number of individuals per infested host).

	Richness	Prevalence	Intensity	
Chaetopterus sp.	1.8	100	3.0	
Chaetopterus cf. appendiculatus	1.02	73.5	1.05	

The distribution pattern of the symbionts among their hosts was primarily regular, as all. O. pettiboneae and most O. fowleri lived solitary inside their host tubes. In turn, Polyonyx cf. heok, Polyonyx sp., E. cometes, Tetrias sp. were found in host tubes usually as male/female pair. The number of nudibranchs varied from one to three per hosts (Table 6) and, in one case, a couple was observed near to a recently spawned egg-mass attached to the inner side of the host tube (Fig. 6H).

The component communities differed also in the relative abundance of a particular species. In the community associated with C. cf. appendiculatus, O. pettiboneae was dominant in terms of both prevalence and abundance. In the community associated with Chaetopterus sp., the dominance of the most abundant symbiont, P. cf. heok is less distinctive, with the role that other species had in the community structure being more relevant (Fig. 7C).

The average length of infested and non-infested tubes of C. appediculatus does not differ significantly (41.2–42.3, t-test, p = 0.65). The number of both symbiotic species and individuals do not show any significant correlation with host tube length in both chaetopterid species.

Discussion

Community dissimilarity

Our results demonstrate a strict segregation in species composition of the communities associated with the two Vietnamese syntopic species of Chaetopterus, which had no species in common. However, at higher taxonomic levels (i.e., family, order and class), they were similar to each other and also resembled the symbiotic communities associated with other species of Chaetopterus and, even, echiuran worms in harbouring scale-worms, pocellanid and pinnotherid decapods and fishes (Anker et al., 2005; Gray, 1961; Ng & Sasekumar, 1993; Petersen & Britayev, 1997).

The two Vietnamese species of Chaetopterus are very similar in body morphology and tube shape, as well as in their trophic-functional characteristics. Thus, no reasons linked to host morphology were evident allowing us to explain the dissimilarity in symbiotic species composition. We suggest therefore that community composition appears to be determined by historical events rather than by the physical or biological habitat characteristics. We may hypothesize that both host species were originally allopatric. Thus, the evolutionary establishment of the respective specialized symbiotic associations would likely occur in different, separated geographical areas, with the posterior disappearance of geographical barriers leading to the current overlapping distribution. Once established, the respective symbiotic communities would be maintained by interspecific competition, leading to symbiont specialization to their respective host species as well as to preventing the exchange of symbionts between hosts when becoming sympatric, even being as closely related as is the case for these two species of Chaetopterus. However, our hypothesis does not exclude the possible existence of differences in host physiology or behaviour that would enhance the ability of the specialized symbionts to compete with possible invaders, thus contributing to maintain the differences in community composition.

Further assessment of this hypothesis would require an experimental approach to analyse the possible existence of a host-factor allowing the respective symbiont to recognize their own hosts, as well as to check the ability of the symbionts from one host to infest the other. In parallel, the regularity of the community segregation would have to be checked by more extensive field sampling addressed to discard (or reveal) the presence of additional symbionts on the alternative host species. This is particularly relevant for Chaetopterus sp. whose accumulation curve (Fig. 7A) supports an expected increase in the respective number of symbiotic species with sampling size. An additional, but not less pertinent question would be to assess the degree of specialization of the symbionts found in the two species of Chaetopterus, either based on previously published data or on our own observations. Therefore, it would be particularly relevant to consider whether they are obligatory or facultative and, in the case of obligatory symbionts, to further assess their degree of specificity (which may range from species-specific to opportunistic).

Symbionts’ specialization

Among the seven species of macroinvertebrates associated with C. cf. appendiculatus and Chaetopterus sp., four (i.e., one polychaete O. pettiboneae, one crab, E. cometes, and one fish, O. fowleri) are known as obligatory symbionts of chaetopterids and other benthic organisms. Onuxodon fowleri also lives in the mantle cavity of bivalves and inside holothurians (Markle & Olney, 1990; Parmentier, Chardon & Vanderwalle, 2002). In our samples, five of six individuals were juveniles, which allows us to suggest that they are employing C. cf. appendiculatus as temporal or intermediate hosts. The porcellanid E. cometes was reported from shallow waters off Singapore, living in association with a species of Chaetopterus, identified as C. variopedatus but this identification is undoubtedly incorrect (Ng & Nakasone, 1993). In turn, the scale-worm O. pettiboneae was first reported from Ambon Island (Indonesia) and later from the coasts of Vietnam, always in association with C. cf. appendiculatus (Britayev & Martin, 2005; Petersen & Britayev, 1997; this paper).

The four other species appear to be undescribed and are now being analysed by the corresponding specialists. However, we may infer some considerations on their degree of specialization based on existing papers dealing with the ecology and morphology of some closely related taxa. Concerning the symbiotic crabs, the porcellanid Polyonyx sp. belongs to the “Polyonyx sinensis” species complex, usually obligatory associates of tubicolous polychaetes, mainly with species of Chaetopterus (B Werding, 2016, unpublished data), while Polyonyx cf. heok belongs to the “Polyonyx pedalis” complex and the same or a very similar species has been recently reported from Singapore living in association with Chaetopterus cf. pacificus (Osawa & Ng, 2016). This suggests that both porcellanids are obligate and, probably, specialized symbionts of Chaetopterus sp. In turn, Tetrias sp. belongs to the Pinnotheridae, a family that mainly includes symbiotic species living as endo- or ectosymbionts in mollusc mantle cavities, polychaete burrows, echinoid integuments or tunicate branchial sacs (Drake et al., 2014). Among them, two species of Tetrias are currently known. Tetrias fischerii (Milne-Edwards, 1867) has been reported as symbiont of bivalves and annelids, while the host of Tetrias scabripes Rathbun, 1898 is unknown (Schmitt, McCain & Davidson, 1973). Although this cannot be assessed from our data, we suggest that the species associated with the Vietnamese Chaetopterus sp. is a specialized obligatory symbiont.

The third undescribed species, the nudibranch Phestilla sp., has several behavioural (i.e., two or more individuals sharing the same host, egg-masses attached to the inner tube surface) and morphological (i.e., posterior end working as a sucker, overall body shape mimicking that of the host) features clearly pointing toward a specialized symbiotic mode of life. This lead us to consider the species as the first know nudibranch living as a symbiont with a marine annelid host. Nudibranchs are well known as aposematic or mimetic organisms (Edmunds, 1987; Gosliner & Behrens, 1989; Rudman, 1991), some of them being considered as true symbionts. Among them, there are some species of Phestilla, which lives in association with corals and are highly specialized predators (Faucci, Toonen & Hadfield, 2007; Robertson, 1970), while the exact nature of the association of the Vietnamese Phestilla sp. and Chaetopterus sp. would need further research to be defined. The single related association occurred between the goniodorid nudibranch Lophodoris scala Marcus & Marcus, 1970 and the innkeeper echiurid Lissomyema exilii (Muüller, 1883). In this case, the nudibranch lives inside the host burrow, sometimes creeping along the host trunk and feeding, possibly exclusively, on Loxosomella spp., an entoproct that colonizes the burrow walls (Ditadi, 1982; Marcus & Marcus, 1970), which seems not to be the case for the Vietnamese species.

Accordingly, all symbionts found in association with Chaetopterus sp. and C. cf. appendiculatus have to be considered as obligatory symbionts. Among them, the less specialized is O. fowleri, which is known to infest hosts belonging to different types of animals (molluscs and polychaetes). The porcellanid crabs E. cometes and P. cf. heok are probably genus-specific symbionts, while the scale-worm O. pettiboneae, together with the other porcellanid crab Polyonyx sp. and the tergipedid nudibranch Phestilla sp., must be considered as species-specific symbionts. The specificity of the pinnotherid crab Tetrias sp. is not clear at this time. However, taking into account the relative abundance of pinnotherids among symbionts of Chaetopterus species (Petersen & Britayev, 1997; Schmitt, McCain & Davidson, 1973), we could also propose that it should be considered as a specialized symbiont, at least at family level.

Therefore, we consider all symbionts found in association with the two Vietnamese species of Chaetopterus as being, or tending to be, specialized symbionts, the single exception being the carapid fish.

Possible causes of observed differences in community structure

We found substantial differences in the structure of the symbiotic communities associated with Chaetopterus sp. and C. cf. appendiculatus. The first shows a significantly higher species richness and mean abundance than the second, while the second was clearly dominated by the presence of a single species, both in terms of abundance and frequency (Table 6, Fig. 6C). Taking into account that body size and tube length of Chaetopterus sp. are significantly lower than those of C. cf. appendiculatus, this situation is particularly unexpected. Usually, species richness and abundance increase with the increasing host size (e.g., Abele & Patton, 1976; Ribeiro, Omena & Muricy, 2003). Thus, the sitiuation of the Vietnamese partnership requires specific considerations.

We suggest that several factors are shaping the differences in the structure of the symbiotic communities associated with Chaetopterus sp. and C. cf. appendiculatus. Despite both host species having low population densities in Nhatrang Bay, that of Chaetopterus sp. was significantly lower, which would likely force the associated symbionts to use (and share) the few available hosts. This would possible explain the higher number of species in its associated community, as well as the fact that all host individuals of Chaetopterus sp. found in Nhatrang Bay harboured symbionts, in contrast to C. appenciculatus whose maximum prevalence was around 70%. Alternatively, the low density of both host populations may impede the secondary dispersion of the symbionts, which has been considered as a key mechanisms shaping the establishment and functioning of marine symbiotic communities (Mekhova et al., 2015) and raises the question on the adults’ ability of long-distance migration.

Based on species and individual’s distributions among hosts, we also hypothesized that another factor determining the observed differences in symbiotic community structure could be the existence of inter- and intraspecific competition. In fact, most tubes of Chaetopterus sp. were occupied by a minimum of two symbiotic species, this co-occurence being very rare in C. cf. appendiculatus, where each host individual was infested by one symbiotic species (Fig. 7B). The single exception was a host tube shared by O. pettiboneae and E. cometes. Accordingly, we suggest that the main driving factors may be resource partitioning between symbiotic species having different sizes, in the case of Chaetopterus sp., and strong interspecific interactions, in the case of C. cf. appendiculatus as previously reported for holothurian hosts (Lyskin & Britayev, 2005).

The characteristics of the symbiotic community structure associated with C. cf. appendiculatus, in which one host was usually occupied by one symbiotic species (Table 5), suggest the existence of interspecific competition between the polychaete and fish symbionts. In turn, the fact that there was a single symbiont per host (Table  5) supports the existence of intraspecific competition among polychaetes and fish individuals, respectively. At least for the polychaete, this hypothesis was supported by our direct observations in experimental aquaria, where individuals of O. pettiboneae were found to fight when trying to occupy the same host tube, as well as by the high frequency of body traumas present (TA Britayev & D Martin, 2016, unpublished data). In contrast, the bulk of Chaetopterus sp. symbionts were crabs (Fig. 7C). Territorial defence is a well-known phenomenon in symbiotic decapods too (Baeza, Stotz & Thiel, 2002; Huber, 1987; Vannini, 1985). However, their behaviour has a sexual component, as they often form heterosexual pairs consisting of gravid males and females co-inhabiting the same host (Castro, 2015; Patton, 1994), which was exactly the case of most porcellanid and pinnotherid crabs inhabiting the tubes of the two Vietnamese species of Chaetopterus (Table 5). This behaviour lead to a significant increase in the abundance of symbiotic individuals in the community associated with Chaetopterus sp. Therefore, while interspecific interactions seemed to affect both species richness and abundance, the intraspecific ones only affected the abundance.

Our observations support two main factors determining the structure of symbiotic communities associated with Chaetopterus sp. and C. cf. appendiculatus in Nhatrang Bay: the density of host populations and competition (both inter- and intraspecific). Moreover, the observed differences in community structure appear to be strictly related to the respective species composition. Accordingly, living in pairs and resource partitioning led to species coexisting in the tubes of Chaetopterus sp. and establish a tightly packed component community showing high species richness and mean intensity, together with a low species dominance. In contrast, the aggressive, strictly territorial species associated with C. cf. appendiculatus established a component community strongly dominated by host being inhabited by a single species and, thus, low species richness and mean intensity.

The existence of two closely related host species with overlapping distributions but harbouring very different symbiotic communities seems to be unusual. We suggest that it may probably be related with the scarcity of data currently available on the structure of symbiotic communities in marine environments. However, the situation is certainly not unique, as at least an additional example has been recently reported from Nhatrang Bay. In this case, the hosts were comatulid crinoids Comanthus gisleni Rowe, Hoggett, Birtles & Vail, 1986 and C. parvicirrus (Müller, 1841) (Mekhova & Britayev, 2012). Consequently, we expect further worldwide studies to discover more syntopic hosts harbouring symbiotic communities with contrasted composition and structure.

Conclusions

Two symbiotic communities inhabit the morphologically similar and syntopic species of the tube-dwelling chaetopterid polychaetes Chaetopterus sp. and C.cf. appendiculatus in Nhatrang Bay. They are mostly composed of specifically specialized species and show a very different composition. The current situation has been attributed to an initially allopatric host distribution allowing the symbiotic communities to be established independently. This is then followed by the subsequent disappearance of the original geographical barriers leading to the current sympatry. The present symbiotic communities differ in structural characteristics (i.e., species richness, mean intensity and species dominance) as a consequence of the differences in host density but also of the existing intra- and interspecific interactions that, in turn, depends on the behaviour of the respective symbiotic species. Mating pairs and partitioned resources lead to a high diversity and intensity in the community associated with Chaetopterus sp., while the aggressive and territorial species associated with C. cf. appendiculatus led to a community with low diversity and intensity but with a strong dominance of a single species.

The hypotheses postulating a similar composition for the symbiotic communities established on closely related hosts seems to reflect a rather common situation in marine ecosystems and, certainly, our results do not allow us to reject it. In fact, the opposite situation was observed within our data, with two taxonomically related hosts living in the same habitat that harbour symbiotic communities with contrasted species composition. Therefore, we hypothesize on the possible reasons explaining their establishment. We also highlight that the situation of the Vietnamese partnerships is certainly not unique and should be considered as an interesting model to further assess different evolutionary and ecological aspects of the establishment of a symbiotic community.

Our results also highlight the importance of studying previously unknown symbiotic associations, which may provide key information allowing the complex network of relationships driving the functioning of the marine ecosystems, particularly in benthic environments, to be understood. Moreover, they are crucial in revealing the hidden biodiversity of the oceans, as supported by the fact that at least five of the nine species herein studied are currently undescribed.

Supplemental Information

Table S1 Main data on the two Chaetopterus hosts

Main data on the two Chaetopterus hosts (tube length in cm, body volume in ml), symbionts’ abundance and number of species, and number of specimens of each symbiotic species found in each individual host specimen tube.

Click here for additional data file.

Table S2 Body volume and tube length of Chaetopterus cf. appendiculatus

Body volume (ml) and tube length (cm) of the 45 specimens of Chaetopterus cf. appendiculatus selected to estimate the relationship between these two measures in the analysis of community structure.

Click here for additional data file.

We would like to thank our colleagues from the Coastal Branch of the Russian-Vietnamese Tropical Centre, Mrs Hai Thanh Nguyen, Mr Tchan Than Kuang and Mr NL Filichev for their help in conducting field and experimental studies, Dr VN Mikheev for fruitful discussion and constructive advices in preparing the manuscript, Dr K Mortimer-Jones for kindly checking the English style of the last version of the manuscript and Dr X Turon for suggesting improvements on the interpretation of the phylogenetic tree. We would also like to thank Professors B Werding and P Ng Kee Lin, and Drs. E Permentier and Irina Ekimova for kindly identifying the porcellanid, pinnotherid, carapid and nudibranch symbionts.

Additional Information and Declarations

Competing Interests

Author Contributions

Field Study Permissions

Data Availability

The authors declare there are no competing interests.

Temir A. Britayev and Daniel Martin conceived and designed the experiments, performed the experiments, analyzed the data, contributed reagents/materials/analysis tools, wrote the paper, prepared figures and/or tables, reviewed drafts of the paper.

Elena Mekhova performed the experiments, analyzed the data, prepared figures and/or tables.

Yury Deart performed the experiments, analyzed the data, prepared figures and/or tables, reviewed drafts of the paper.

The following information was supplied relating to field study approvals (i.e., approving body and any reference numbers):

Russian–Vietnamese Tropical Center approved the collection of the samples and animals used in our article.

The following information was supplied regarding data availability:

ResearchGate: https://www.researchgate.net/publication/310159685_Phestilla_sp.

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
