# Peer review of "Do syntopic host species harbour similar symbiotic communities? The case of Chaetopterus spp. (Annelida: Chaetopteridae)"

_PeerJ, doi:10.7717/peerj.2930_

## Round 0.1 · original submission · Major Revisions

· Academic Editor

Major Revisions

I have now heard back from two reviewers. While both were positive about your submission, reviewer 1 has some questions about the identity of the chaetopterid species. Please examine this point in detail before any resubmission, along with the other helpful suggestions by both reviewers.

Reviewer 1 ·

Basic reporting

This paper compares the host symbiont composition in two sympatric Chaetopterus species. The authors examined the association between the hosts and their symbionts in four localities off the coast of Vietnam, the results could contribute to a better understanding of how hosts can modify the marine biodiversity. The paper is in general well-written, with details of data shown. However, the authors may need to address the critical question of species diversity of the hosts. It is unusal for two closely related species of Chaetopterus to live sympatrically. Therefore the authors need to provide more evidence to convince the readers they are actually two species. I will elaborate this below.

Experimental design

In general good, but again I have some doubt about the idemtity of the two host species. If these worms turns out to be the same species, the authors may need to rewrite the whole manuscript and seek other possible xplanation for the differential symbiont composition.

Specifically, the type locality of Chaetopterus appendiculatus is Ceylon in western Indian Ocean, which is very far away from Vietnam in the Pacific Ocean. Many Chaetopterus species that were considered to have a wide distribution turn out to be different species. Furthermore, the authors may have made a mistake in stating the denticles of the neuropodial uncini of region C for this speies is 25-35 (L. 163) because in all species of this genus the number is less than 10.

For Chaetoperus sp.: The authors only listed several differences between this species and C. appendiculatus: 1. size; 2, number of chaetigers in region A (9 vs. 9-11); 3, denticles of uncini in region C (9 vs 25-35). But the body size can vary according to stage of maturity and the habitat characteristics, 9 is not well-separated from 9-11 in terms of teh number of chaetogers in region A. I have question the validity of point 3 earlier.

My suggestions are to provide a more detailed comaprison of the morphology of the "two species", including providing photos of the uncini, and to conduct some molecular comparisons, given that these worms are very common and their is no problem with obtaining etanol preserved specimens. If in deed the uncini denticles are 25-35 vs. 9, I would trust that they are different species. For molecular analysis, they could consider to amplify the COI, 18S and 28S gene fragments and compare them with sequences available in GenBank, conduct a phylogenetic analysis to see whether these "two species" are actually different species, and what is the most closely related species to these two species. These gene fragments have been used by Dr Martin (the corresponding author) in a previous study of animals in Chaetopteridae (Martin, D.G.J., Carreras-Carbonell, J. & Bhaud, M. (2008) Description of a new species of Mesochaetopterus (Annelida, Polychaeta, Chaetopteridae), with redescription of Mesochaetopterus xerecus and an approach to the phylogeny of the
family. Zoological Journal of the Linnean Society, 152, 201–225). I wonder why they did not do the same for these Chaetopterus.

Validity of the findings

The identities of the symbionts have been verified by experts of the respective groups. I have no doubt about them. The authors themselves are experts in polychaetes. However, i would urge them to pay more attention to the identities of the polychaete hosts.

Additional comments

Please see my suggestions regarding the identities of the hosts.

Reviewer 2 ·

Basic reporting

I think this article report contrasting symbiotic community in syntopic and closely related species hosts. This finding is useful for understanding the symbiotic community mechanism. Therefore, there is no reason for reject. However, I found some minor revision of this article (please see the word file).
I wish more assessment about the relationship between host tube structure/host physiological trait and symbiotic species in the future.

Experimental design

No Comments

Validity of the findings

No Comments

Annotated reviews are not available for download in order to protect the identity of reviewers who chose to remain anonymous.

---

## Round 0.2 · Minor Revisions

· Academic Editor

Minor Revisions

Overview: While both reviewers now feel the paper is ready to be published, I have looked over the paper in detail myself, and have three major points that need to be addressed before final acceptance.

1. The English in many areas needs much editing, and there are many mistakes including simple grammatical errors. I have edited the Abstract and Introduction to give you an idea of how much work is needed, but I have not touched most of the remaining text. Please have someone thoroughly go over the paper who is a native English speaking trained biologist, and please provide their name in your rebuttal comments. As stated in the PeerJ Instructions to Authors, English is the responsibility of the authors, not reviewers or the editor.
2. The results section explaining Figure S1 needs much more information. The results strongly show the presence of two species, even if the overall phylogeny of the group is not clear. Please add information as requested in the attached PDF file, and consider moving the figure to the main paper from Supp. Materials.
3. Finally, the species name you give for one species (cf. appendiculatus) is not accepted in the World Register of Marine Species. You must provide reasoning or references for why you are using this name in the paper. Also, the naming of the species is not consistent in the text and figures. All of these taxonomic issues must be exactly resolved as you are dealing with some undescribed and uncertain species. As well, avoid the term “new species” please; this has special taxonomic meaning for original descriptions.

There are some other small comments in the attached PDF, and I look forward to seeing a revised version.

Reviewer 1 ·

Basic reporting

The authors have addressed my concerns raised in the previous run of review. In my opinion, the manuscript is ready for publication.

Experimental design

I have no additional comments on teh experimental design

Validity of the findings

The findigs look valid to me.

---

## Round 0.3 · Minor Revisions

· Academic Editor

Minor Revisions

The paper is very well revised, with only some small edits needed before acceptance; hence my decision is "minor revisions". Please take a look at the attached pdf.

Note there are two pdf files - and I can only attach one here, so I will send the other one to you via normal e-mail.

---

## Round 0.4 · accepted · Accept

· Academic Editor

Accept

Thank you very much for your hard work in the revisions - and I look forward to seeing the published version of your work!